# Targeted Combination Antibiotic Therapy Induces Remission in Treatment-Naïve Crohn’s Disease: A Case Series

**DOI:** 10.3390/microorganisms8030371

**Published:** 2020-03-06

**Authors:** Gaurav Agrawal, Annabel Clancy, Rijata Sharma, Roy Huynh, Sanjay Ramrakha, Thomas Borody

**Affiliations:** Centre for Digestive Diseases, 1/229 Great North Rd, Five Dock, NSW 2046, Australia; Annabel.Clancy@cdd.com.au (A.C.); rijata.sharma@gmail.com (R.S.); roy.huynh@live.com (R.H.); ramrakha.sanjay@gmail.com (S.R.); Thomas.Borody@cdd.com.au (T.B.)

**Keywords:** anti-bacterial agents, Crohn’s disease, inflammatory bowel disease, mucosal healing, *Mycobacterium avium* subspecies *paratuberculosis*

## Abstract

Prospective trials of anti-mycobacterial antibiotic therapy (AMAT) have proven efficacious in Crohn’s disease (CD) but use as first-line treatment in CD has not been evaluated. This paper reports the outcomes of patients with CD treated with first-line AMAT. This paper consists of a case series of treatment-naïve CD patients who received AMAT as first-line treatment between 2007 and 2014 at a single center. AMAT treatment consisted of rifabutin, clofazimine and clarithromycin, plus either ciprofloxacin, metronidazole or ethambutol. Symptoms, inflammatory blood markers, colonoscopy and histology results, in addition to, the Crohn’s Disease Activity Index (CDAI) were tabulated from patients’ clinical records, and descriptive statistics were conducted. A Wilcoxon signed-rank test assessed the difference in CDAI scores before and while on AMAT. The statistical significance was set at 5%. Clinical remission (CDAI < 150) with rapid improvement in clinical symptoms and inflammatory markers was seen in all eight patients receiving AMAT as sole therapy by 6 weeks. In all eight patients, the median CDAI score decreased significantly, from 289 prior to treatment to 62 at the 12-month follow-up (*p* < 0.001). Follow-up colonoscopies showed healing of CD ulcers, no visible mucosal inflammation, restoration of normal vascular patterns and complete mucosal healing on histology samples. AMAT as first-line therapy demonstrated a rapid improvement of Crohn’s disease (not previously seen when used as second-line therapy).

## 1. Introduction

Anti-inflammatory and immunosuppressive agents have evolved to be the standard of care for Crohn’s disease (CD) [1]. A number of biological agents targeting various cytokines and components of the inflammatory cascade have been approved for CD [2,3], expanding anti-inflammatory options for patients. However, despite these advances, more than 60% of patients fail to achieve remission by six weeks and roughly 50% fail to achieve remission by one year [2]. In addition, a further 23–46% of patients experience a loss of response over time (termed secondary non-responders) [4,5]. Although dampening the immune response has delivered modest results, these findings tell us the underlying cause of the disease persists unaddressed by current therapies, and newer therapies are needed.

Antibiotic therapies have long been used for the treatment of CD [6] and aim to target potential pathogenic agents which have been associated with CD and the imbalance in the gut microbiome commonly seen in patients with CD [7]. One candidate pathogen is *Mycobacterium avium* subspecies *paratuberculosis* (MAP), which is the causative organism of a near-identical inflammatory bowel disease in ruminants and primates, termed ‘‘Johne’s disease’’ [6]. It belongs to the *Mycobacterium avium* complex (MAC) group, one of many non-tuberculous mycobacteria (NTM), which are distinct from the other mycobacteria group named the *Mycobacterium tuberculosis* complex (MTBC), of which some cause pulmonary tuberculosis.

Members of the MAC group are considered less pathogenic (more opportunistic) with different behaviors than those seen with MTBC. There has been a trend of an increased rate of non-tuberculosis mycobacteria (NTM) infection diagnoses in recent years [8]. This is due to better clinical awareness and recognition of presentations and improved laboratory diagnostic methods in immunocompromised and immunocompetent patients. This problem of reliable diagnostics is also seen with MAP, although these same developments are being implemented in its detection. Culture with the Ziehl–Neelsen stain is not appropriate and there are questions as to the suitability of current PCR techniques, which test for the presence of MAP only and have been found to be in a high percentage of controls. Implementing optimum PCR techniques/processing has shown more accuracy in MAP detection in Crohn’s patients [9]. In addition, there are promising culture-based [9,10] and flow cytometry methods in development [11].

Studies on antibiotic therapy targeting MAP (i.e., anti-mycobacterial antibiotic therapy, AMAT) have been mixed, with results predominantly coming from observational, open label or case series studies. Several prospective trials of AMAT targeting atypical mycobacteria have proven efficacious in CD, with clinical remission rates reaching ~93% [12,13,14]. The first large, successful randomized control trial (RCT) of AMAT for CD, on an intention-to-treat re-analysis for statistical errors, showed significant induction and maintenance of remission [15]. This is an important distinction, as atypical mycobacteria are known to be more difficult to eradicate than the MTBC, which reaches a cure rate of 81% for fully sensitive organisms (World Health Organization Data 2018) [16]. This re-evaluated finding was recently confirmed by a larger international, Phase III RCT (NCT01951326) in 331 CD patients, which met its primary endpoint of clinical remission, defined as a Crohn’s Disease Activity Index (CDAI) < 150 at week 26 (*p* = 0.007). Secondary endpoints were also successfully met including clinical response at week 26 (*p* = 0.016), early clinical remission at week 16 (*p* = 0.015), clinical remission at weeks 16 and 52 (*p* = 0.003) and durable remission at all visits through to week 52 (*p* = 0.018) [17]. Importantly, the majority of these studies utilized AMAT as a three-drug combination of rifamycin, macrolide and clofazimine, as the assumption is that MAP behaves much the same as other NTM (hence, isoniazid and pyrazinamide are thought to be suboptimal). However, depending on the severity of the disease, NTM require four drugs [18] but data are limited, mostly from single-center observational studies and expert opinions. Factors that need to be taken into consideration are that MAP is more pathogenic, resistant and has the slowest growth rate of all NTM. Additionally, individual drug effectiveness can differ [19], especially between different drug generations, and so a four-drug combination may be more optimal, even if there is a “step down” approach after a number of months. This fourth drug used depends on specific patient factors, such as site of disease, allergies, presence of fistula, prior therapy, concomitant illnesses, etc., and is representative of “personalized patient treatment.” 

In the past, all patients treated with AMAT have been on standard anti-inflammatory drugs before being referred for AMAT (AMAT as second-line therapy). Here, we report for the first time a case series of treatment-naïve CD patients treated with AMAT as first-line therapy. 

## 2. Materials and Methods

Treatment-naïve CD patients who requested off-label AMAT as first-line treatment in lieu of standard immunosuppressive treatment between December 2007 and December 2014 were identified from the medical records database. This AMAT was comprised of four drugs, with the fourth choice selected by the attending gastroenterologist depending on allergies, fistula or site of disease. 

Patients were consecutively included if they had a confirmed diagnosis of CD (clinical, endoscopic and histopathological diagnostic features), had not previously received any treatment for CD and received combination antibiotic therapy for CD. Exclusion criteria were unconfirmed CD or other inflammatory bowel diseases and previous treatment for CD including enteral nutrition, biologics or anti-inflammatories.

In December 2018, data were tabulated in Microsoft Excel from patients’ clinical records at baseline, 6 weeks and 12 months. Data included demographics, symptoms, inflammatory blood markers (C-reactive protein, erythrocyte sedimentation rate and hemoglobin), colonoscopy and histology results, and the CDAI calculated at the 12-month follow-up visits. Results from the colonoscopy biopsies were examined to document the presence of “complete mucosal healing,” defined as the restoration of normal mucosal appearance, with the absence of mucosal ulcerations and of histological evidence of inflammation at available patient follow-up dates [20,21]. Concomitant therapies and adverse events were also recorded. Descriptive statistics were conducted, and a Wilcoxon signed-rank test was used to assess the difference in CDAI scores before and whilst on AMAT. The statistical significance was set at 5%. This study was approved by the institutional ethics committee (CDD19/C01). The statistical review of the study was performed by a biomedical statistician.

## 3. Results

Eight patients (4 males; 4 females; aged 12–46 years) were identified to meet the inclusion criteria. All eight patients were newly diagnosed with CD at our clinic (single center, day hospital) and received extensive counselling regarding available therapeutic options. Patients declined immunosuppressive medications and instead chose AMAT. After obtaining informed consent, AMAT was commenced in a dose-escalating fashion over an average of eight weeks or more to reach final doses of rifabutin (4 mg/kg), clofazimine (3.75 mg/kg) and clarithromycin (14 mg/kg). Five of the eight patients were also on ciprofloxacin (12 mg/kg) and/or metronidazole (7.5 mg/kg).

Improvement in clinical symptoms and inflammatory markers was achieved in all eight patients receiving AMAT as sole therapy by 6 weeks. In all eight patients, the median CDAI score decreased significantly, from 289 prior to treatment to 62 at the 12-month follow-up (*p* < 0.001). Clinical remission (CDAI < 150) was achieved in all patients (Table 1). 

At the 12-month follow-up colonoscopy (*n* = 7), there was no evidence of contact bleeding or visible mucosal inflammation, with restoration of normal vascular patterns. Healing of CD ulcers was observed, with no surrounding inflammation (Figure 1b). Complete mucosal healing was confirmed by histology results. Comparison of pre-treatment and response-to-treatment blood results showed resolution of anemia in all patients, increased hemoglobin and ferritin levels and a decrease in inflammatory markers to within normal ranges.

Adverse effects were minimal for the duration of the follow-up. Liver function was monitored in all patients. Patient 5 experienced an elevation in liver function tests (LFTs), which was managed by the substitution of rifabutin with rifampicin. Reversible skin discoloration resembling a suntan and orange urine discoloration, a known side effect of clofazimine/rifabutin, were observed in all patients. Three patients reported transient arthralgia (duration of four weeks to three months).

AMAT-induced healing proved to be long-term. One patient ceased the therapeutic regimen after eight years of continuous use and has remained clinically, endoscopically and histologically well for eight years as of 2018. Two patients, who initially experienced a complete resolution of CD, independently ceased AMAT, then relapsed after two years off treatment. These patients later resumed low-dose maintenance AMAT and regained remission within weeks. After achieving remission, clinical, hematological and biochemical findings remained within normal ranges for a period of ≥3 years.

## 4. Discussion

To our knowledge, this is the first case series documenting successful induction of remission of CD symptoms and mucosal inflammation using AMAT as the first-line therapy in treatment-naïve CD patients. Induction of remission with reduction in the CDAI was achieved in all patients who received AMAT as sole initial therapy. Adverse effects with AMAT were minimal and included reversible urine and skin discoloration in all patients and transient arthralgia in 3/8 patients. 

The key findings of our study were: (a) induction of remission; (b) remission in all patients; and (c) complete mucosal healing. Remission induction with AMAT (reported previously) took much longer (>16–52 weeks) than the six-week follow-up (4–21 weeks) reported here, and only 60–80% reached remission [22]—unlike the 100% achieved here—albeit in a small cohort. These superior results in treatment-naïve CD may in part be explained by the lack of previous exposure to anti-inflammatory and immunomodulatory agents. The long-term healing observed even after stopping antibiotics, a finding unmatched by immunomodulators, negates the notion that antibiotic-related immune modulation contributed to our results. In summary, antibiotics targeting AMAT, used alone, induced remission of CD. Reliable and accurate diagnostics are urgently needed to confirm this observation and determine whether they are acting on MAP specifically or on other bacteria. This case series makes a strong case for urgent randomized control trials (RCTs) to investigate this effect further and whether AMAT should be used as first-line therapy for CD to achieve remission.

Remarkable similarities have long been observed between the clinical features and gross pathology of CD and Johne’s disease in ruminants, caused by MAP [6]. Since these initial observations, a wealth of data from epidemiological [23,24,25], genetic [26,27,28], experimental human and animal studies [29], clinical trials [12,17,30] and meta-analyses [31,32] have converged to establish a potential pathogenic role for MAP in CD. Yet despite these associations between CD and this well-recognized zoonotic pathogen, the role of MAP in CD remains contentious, in part due to the lack of availability of a diagnostic test [17,33]. Despite the lack of testing for MAP in patients in this study, a number of unique characteristics of this treatment-naïve cohort provide support for MAP as an infective causative agent of CD. For example, infliximab [34] and many of the anti-inflammatory and immunomodulatory drugs effective in CD [35,36,37,38] have been shown to possess weak anti-MAP activity. Thus, prior exposure to these may equate to suboptimal antibiotic exposure leading to MAP resistance, perhaps promoting intracellular dormancy and loss of AMAT efficacy. Relapse in two of our patients after a long remission points to the reactivation of dormant or cell-wall-deficient MAP, a shared characteristic of mycobacteria, which also supports this view. Once accurate diagnostic testing is available and can include sensitivities of the organism to drugs, this theory can be confirmed, and treatment regimens can be optimized and focused. 

“Treatment-naïve” CD patients are generally diagnosed in hospitals, increasingly by pediatric gastroenterologists. This study was limited due to patients being recruited from a tertiary adult outpatient center. A small number of newly diagnosed/ “treatment-naïve” presenting CD patients were seen who sought AMAT treatment, representing the small cohort collected. The small sample size and inherent limitations of case series studies are recognized. Efforts were made to reduce bias including a clear study objective, explicit inclusion/exclusion criteria, specified time for patient recruitment, consecutive patient enrolment, clinically relevant outcomes, prospective data collection and a high follow-up rate. Once accurate diagnostic testing for MAP infections of humans is available, especially if it includes antimicrobial sensitivity testing of MAP isolates, this theory can be confirmed and treatment regimens better optimized.

A key point is whether the AMAT drugs used are specific for MAP or work by general targeting of the gut microbiome. However, this is likely to be both, as the medications act intracellularly and some, such as clofazimine, are specific for mycobacteria [39,40]. 

Given the severe impact of CD on patients and their families [41,42], incomplete responses [43,44] and associated risks with immunosuppressive medications [45,46] versus the favorable side effect profile of AMAT [17,30], a trial of AMAT versus existing therapies should be initiated in pediatric CD. Furthermore, these results suggest that AMAT may end up being used as first-line therapy in treatment-naïve patients for all forms of CD including the severe category, reserving immunomodulators and biologics as rescue medications.

## 5. Conclusions

In conclusion, we report for the first time remission of CD with deep mucosal healing in treatment-naïve patients using AMAT. Although preliminary, these striking observations warrant randomized controlled trials of AMAT as first-line therapy for “treatment-naïve” CD patients versus those who have failed previous treatments. The results of the completed RHB-104 Phase III AMAT study (NCT01951326) [17] underscore the efficacy of AMAT in CD. Our results dictate that future trials also examine AMAT as a first-line rather than rescue therapy. In addition, a four-drug combination is thought to be optimal, and AMAT’s decreased side effect profile and omission of immune suppression therapy make it a favorable option for a future pediatric study. 

## Figures and Tables

**Figure 1 microorganisms-08-00371-f001:**
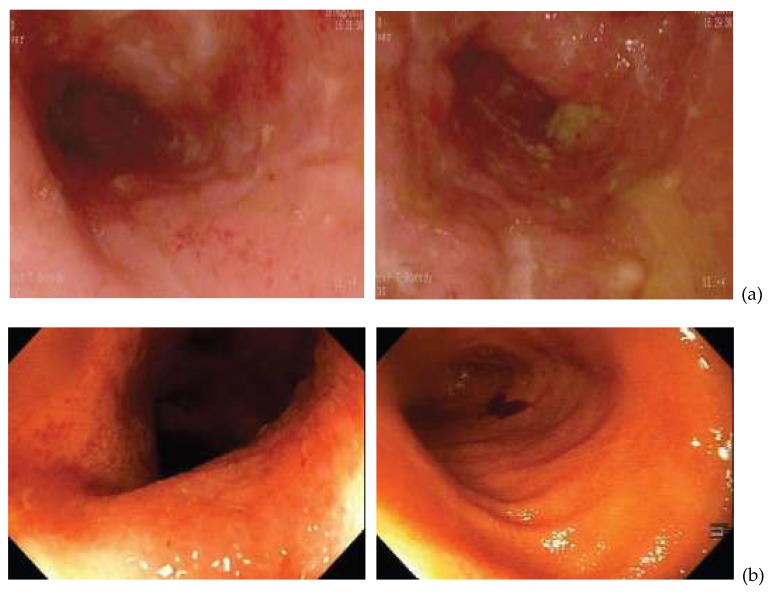
Improvement in terminal ileum inflammation following anti-mycobacterial antibiotic therapy (AMAT). (**a**) Patient 1: Initial view of inflamed, ulcerated terminal ileum (2011). (**b**) Patient 1: Follow-up view of the terminal ileum (2012). Healed mucosa showing healthy villi with some red clofazimine staining spots in the distal ileum.

**Table 1 microorganisms-08-00371-t001:** Summary of Clinical and Biochemical Response to AMAT.

Pt.	Demographics (Gender, Age)	Baseline			Treatment	6 Weeks		12 Months		
		Symptoms	CDAI	CRP		Symptoms	CRP (mg/L)	Symptoms	CDAI	CRP (mg/L)
1	F, 30	BM 10/day, abdominal pain	367	11.3	Rifabutin, clarithromycin, clofazimine, ciprofloxacin	BM 2–3/day	10.5	BM 2–3/day, no abdominal pain	85	5.9
2	M, 29	Abdominal pain, weight loss	230	142.9	Rifabutin, clarithromycin, clofazimine, metronidazole	BM 2–3/day	8.1	BM 2–3/day	31	2.1
3	F, 35	BM 5–6/day, abdominal pain, weight loss	346	32	Rifabutin, clarithromycin, clofazimine, metronidazole	BM 1–2/day	3.8	BM 1/day	16	1.0
4	M, 27	Abdominal pain, weight loss	298	68	Rifabutin, clarithromycin, clofazimine, ciprofloxacin	BM 1–3/day, no abdominal pain	2.8	BM 2/day	78	0.5
5	M, 46	BM 2–6/day, weight loss	269	5.7	Rifabutin, clarithromycin, clofazimine, metronidazole	BM 2–3/day	9.6	BM 2–3/day	87	1.7
6	M, 13	BM 5/day, abdominal pain, weight loss	468	42.6	Rifabutin, clarithromycin, clofazimine	BM 1–2/day, reduced abdominal pain, weight gain 2 kg	12.1	BM 2–4/day	88	3.5
7	F, 14	BM 3–4/day	273	5.7	Rifabutin, clarithromycin, clofazimine, ciprofloxacin	BM 2–3/day	1.7	BM 3/day	45	0.4
8	F, 19	BM 20/day, abdominal pain	280	10	Rifabutin, clarithromycin, clofazimine, metronidazole	BM 2/day, no pain	3.9	BM 2/day	46	4.0

CDAI: Crohn’s Disease Activity Index (normal <150); CRP: C-reactive protein (normal 0–6); BM: Bowel Motion(s).

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
