# Peer review of "Targeted Combination Antibiotic Therapy Induces Remission in Treatment-Naïve Crohn’s Disease: A Case Series"

_microorganisms, 2020, doi:10.3390/microorganisms8030371_

Round 1

Reviewer 1 Report

This report is novel and a major contribution to the field of paratuberculosis as a zoonotic pathogen.  The following minor edits are intended to strengthen the work, listed by line number.

12           The statement begins: “Single centre case….” is not a complete sentence.

40           Change “a” to “the” as in “MAP is the causative organism”.

41           “Johne’s disease” is not fully in quotation marks and the “d” is normally not capitalized.  Also, remove the uncertainty in this sentence such that it reads: “It belongs to the Mycobacterium avium Complex (MAC), one of many non-tuberculous mycobacteria (NTM) which are distinct from mycobacteria in the Mycobacterium tuberculosis Complex (MTBC).”  

42           The word “complex” should not be italicized.

43           This sentence, beginning “It is currently thought … “  is poorly written and can be deleted.

45           Missing reference.

47           The sentence beginning “It is unclear….” makes no sense and should be re-written.  I do think that the lack of validated diagnostic tests for MAP in humans should be mentioned in the introduction of this manuscript, but this sentence fails to do this in a meaningful way.

50           Suggested revision: “…targeting MAP, i.e. anti-mycobacterial therapy (AMAT), have been…”

55           Suggested revision: “This is an important distinction, mycobacteria in the MAC are notoriously difficult to treat.  Even for more susceptible mycobacteria like those in the MTBC, cure rates of only 81% are the norm (World Health Organization Data 2017).”

63           Do not capitalize the words rifamycin or macrolide or clofazimine.

66           Change to read: “NTMs require….” (delete the “s” in require).  The word “data” is plural and thus the sentence should read “data are…”.

68           Change “longest reproductive rate” to “slowest growth rate of all NTMs.”

115         Anemia is mentioned here, and hemoglobin measurements are described in the methods section (see line 88) but no such results are in Table 1.  Likewise, ESR is mentioned in the methods (line 88) but no data are found in Table 1.  Likewise, ferritin levels are mentioned on line 116 but no results are found in Table 1.  So, either fix table 1 or describe the results in the text.

124         The acronym “LFTs” is not defined.

148         Suggested revision: “In summary, AMAT therapy alone induced, and later re-induced, remission of CD.”

149         The acronym “RCTs” is not defined.

159         Hyphenate “treatment-naïve” as you did elsewhere.

165         Suggested revision: “Once accurate diagnostic testing is for MAP infections of humans is available, especially if it can include antimicrobial sensitivity testing of MAP isolates, this theory can be confirmed and treatment regimens better optimized.”

170         The meaning of the sentence beginning “Once a few….” Is unclear and should be re-written.

179         Do not capitalize clofazamine.

180         Suggested revision: “Given the severe impact of CD on patients and their families,…..”

All references should all be reviewed for accuracy. 

Examples of possible errors, by ref number:

10&11   Is LIVER supposed to be in all caps?

13           Parallel is misspelled

20-36     Bacterial names are not italicized as they should be.

22           A portion of the article title seems missing.

25           Insert space between “a” and Mycobacterium (which should also be italicized)

30           The end of the article title seems in error (why pdf ???)

Author Response

Thank you to the reviewer for his time and comments. They have all been addressed:

12           The statement begins: “Single centre case….” is not a complete sentence. Amended

40           Change “a” to “the” as in “MAP is the causative organism”. Amended

41           “Johne’s disease” is not fully in quotation marks and the “d” is normally not capitalized.  Also, remove the uncertainty in this sentence such that it reads: “It belongs to the Mycobacterium avium Complex (MAC), one of many non-tuberculous mycobacteria (NTM) which are distinct from mycobacteria in the Mycobacterium tuberculosis Complex (MTBC).”   Amended

42           The word “complex” should not be italicized. Amended

43           This sentence, beginning “It is currently thought … “  is poorly written and can be deleted. Amended

45           Missing reference. included

47           The sentence beginning “It is unclear….” makes no sense and should be re-written.  I do think that the lack of validated diagnostic tests for MAP in humans should be mentioned in the introduction of this manuscript, but this sentence fails to do this in a meaningful way. Amended

50           Suggested revision: “…targeting MAP, i.e. anti-mycobacterial therapy (AMAT), have been…” Amended

55           Suggested revision: “This is an important distinction, mycobacteria in the MAC are notoriously difficult to treat.  Even for more susceptible mycobacteria like those in the MTBC, cure rates of only 81% are the norm (World Health Organization Data 2017).” Amended

63           Do not capitalize the words rifamycin or macrolide or clofazimine. Amended

66           Change to read: “NTMs require….” (delete the “s” in require).  The word “data” is plural and thus the sentence should read “data are…”. Amended

68           Change “longest reproductive rate” to “slowest growth rate of all NTMs.” Amended

115         Anemia is mentioned here, and hemoglobin measurements are described in the methods section (see line 88) but no such results are in Table 1.  Likewise, ESR is mentioned in the methods (line 88) but no data are found in Table 1.  Likewise, ferritin levels are mentioned on line 116 but no results are found in Table 1.  So, either fix table 1 or describe the results in the text. Amended

124         The acronym “LFTs” is not defined. Amended

148         Suggested revision: “In summary, AMAT therapy alone induced, and later re-induced, remission of CD.” Amended

149         The acronym “RCTs” is not defined. Amended

159         Hyphenate “treatment-naïve” as you did elsewhere. Amended

165         Suggested revision: “Once accurate diagnostic testing is for MAP infections of humans is available, especially if it can include antimicrobial sensitivity testing of MAP isolates, this theory can be confirmed and treatment regimens better optimized.” Amended

170         The meaning of the sentence beginning “Once a few….” Is unclear and should be re-written. Amended

179         Do not capitalize clofazamine. Amended

180         Suggested revision: “Given the severe impact of CD on patients and their families,…..” Amended

All references should all be reviewed for accuracy. 

10&11   Is LIVER supposed to be in all caps? Amended

13           Parallel is misspelled Amended

20-36     Bacterial names are not italicized as they should be. Amended

22           A portion of the article title seems missing. Amended

25           Insert space between “a” and Mycobacterium (which should also be italicized) Amended

30           The end of the article title seems in error (why pdf ???) Amended

Reviewer 2 Report

Authors present a high-quality experimental manuscript that describes targeted combination antibiotic therapy that induces remission in treatment-naïve Crohn’s Disease. Authors report for the first time the outcomes of patients with CD treated with first-line anti-mycobacterial antibiotic therapy (AMAT). The manuscript is well-written and a pleasure to read. Authors observed that for 8 patients the median Crohn’s Disease Activity Index (CDAI) score decreased significantly from 289 prior to treatment to 62 at the 12 month follow up. Based on the results, authors conclude that: - Improvement in clinical symptoms and inflammatory markers was achieved in all eight patients receiving AMAT as sole therapy by 6 weeks. - AMAT as first-line therapy can demonstrate a rapid improvement of Crohn’s Disease. - Clinical remission (CDAI

Author Response

Thank you to the Reviewer's comments and feedback. We have addressed the English and reviewed and updated the results section.

Sincerely,